# Some Special Aspects of Liver Repair after Resection and Administration of Multipotent Stromal Cells in Experiment

**DOI:** 10.3390/life11010066

**Published:** 2021-01-18

**Authors:** Igor Maiborodin, Elena Lushnikova, Marina Klinnikova, Swetlana Klochkova

**Affiliations:** 1Laboratory of Cell Biology and Cytology, Institute of Molecular Pathology and Pathomorphology, Federal State Budget Scientific Institution “Federal Research Center of Fundamental and Translational Medicine” of the Ministry of Science and Higher Education of the Russian Federation, Acad. Timakov st., 2, 630117 Novosibirsk, Russia; pathol@inbox.ru (E.L.); margen@ngs.ru (M.K.); 2Laboratory of Health Management Technologies, The Center of New Medical Technologies, Institute of Chemical Biology and Fundamental Medicine, The Russian Academy of Sciences, Siberian Branch, Akademika Lavrenteva str. 8, 630090 Novosibirsk, Russia; 3Department of Human Anatomy, Peoples Friendship University of Russia (RUDN University), Miklukho-Maklaya st., 117198 Moscow, Russia; swetlana.chava@yandex.ru

**Keywords:** multipotent mesenchymal stromal cells, liver, liver resection, neutrophils, macrophages

## Abstract

Changes in rat liver after resection and injection of autologous multipotent mesenchymal stromal cells of bone marrow origin (MSCs) transfected with the GFP gene and cell membranes stained with red-fluorescent lipophilic membrane dye were studied by light microscopy. It was found that after the introduction of MSCs into the damaged liver, their differentiation into any cells was not found. However, under the conditions of MSCs use, the number of neutrophils in the parenchyma normalizes earlier, and necrosis and hemorrhages disappear more quickly. It was concluded that the use of MSCs at liver resection for the rapid restoration of an organ is inappropriate, since the injected cells in vivo do not differentiate either into hepatocytes, into epithelial cells of bile capillaries, into endotheliocytes and pericytes of the vascular membranes, into fibroblasts of the scar or other connective tissue structures, or into any other cells present in the liver.

## 1. Introduction

The safety and success of donor liver recovery is a top priority in organ transplantation [1,2]. Besides that, the liver is resected within healthy tissues in cases of extensive traumatic damage [3,4], during oncological processes [5,6], and the development of metastases [7,8], upon detection of significant benign tumors [9,10] and for the elimination of some parasitic invasions, such as echinococcosis [11,12] and alveococcosis [13,14].

Transplantation of multipotent stromal cells (MSCs) is a new factor in the correction of liver failure that developed after the removal of a large fragment of the organ. It is traditionally recognized that in liver pathology, MSCs exert their therapeutic effect, mainly through transdifferentiation into hepatocytes and nonparenchymal cells of the liver [15,16]. Additionally, MSCs in liver failure, due to ischemic and reperfusion injuries before transdifferentiation, secrete various trophic and immunomodulatory factors that reduce the severity of pathological changes and decrease the activity of the inflammatory response [17,18].

Along with this, there is evidence in the literature that after injection into tissues, the MSCs die very quickly due to a sharp change in the microenvironment [19,20,21,22,23,24]. In addition, all works devoted to the effect of MSCs and their exosomes on the liver and reporting a good therapeutic efficacy of cell therapy do not contain any data on the possible complications and side effects of MSCs.

In connection with the above, the aim of the study was set — to evaluate the results of using autologous mesenchymal MSCs of bone marrow origin (MMSCs) to influence the repair of the resected liver in the experiment.

## 2. Material and Methods

The work is based on the results of a study of the liver of male inbred Wag line rats weighing 180–200 g and aged 6 months at different times after the injection of MMSCs into the site of organ resection. The manipulations did not cause pain to the animals and were carried out in compliance with Russian legislation: GOST 33215-2014 (Guidelines for accommodation and care of laboratory animals. Rules for equipment of premises and organization of procedures) and GOST 33216-2014 (Guidelines for accommodation and care of laboratory animals. Rules for the accommodation and care of laboratory rodents and rabbits). The work was approved by the Committee for Biomedical Ethics (20 November 2020, decision № 32) of a Federal State Budget Scientific Institution “Federal Research Center of Fundamental and Translational Medicine” (FRC FTM).

### 2.1. Preparation, Cultivation, and Characteristics of MMSCs

MMSCs were obtained from the bone marrow of a male Wag inbred line rat weighing 180 g and aged 6 months, and then were characterized and cultured as described in our previous works [21,22,25,26]. Isolated MMSCs express some characteristic MSC markers (CD73, CD90, and CD105) and do not express the hematopoietic markers (CD14, CD20, CD45, and CD34).

The capacity of induced in vitro differentiation into osteogenic, adipogenic, and chondrogenic lineages is the only critical requirement for choosing MSCs population. MSCs can differentiate into bone tissue cells, and the conditions of this differentiation can be easily reproduced; therefore, this differentiation is routinely used for in vitro characterization of MSCs cultures and is a typical default differentiation pathway for the majority of MSCs in culture. Osteogenic differentiation was induced by adding 0.1 μM desoxymethasone, 50 μM ascorbic acid, and 10 μM β-glycerophosphate (Sigma) [27,28,29]. Osteogenic differentiation was determined by two markers: activity of alkaline phosphatase and mineralization of extracellular matrix with calcium ions. Cytochemically, alkaline phosphatase was detected using nitroblue tetrasolium in the presence of 5-bromo-4-chloro-3-indolyl phosphate as the substrate. Accumulation of calcium in extracellular matrix was recorded by alizarin red staining.

MMSCs of the 2nd passage were transfected with the DNA of the plasmid pEGFP-N1 (Clontech Laboratories Inc., Mountain View, CA, USA) containing the gene of the green fluorescent protein GFP. Additionally, MMSCs were stained with red-fluorescent lipophilic membrane dye the Vybrant^®^ CM-Dil (Thermo Fisher Scientific, Waltham, Massachusetts, USA), which binds to the membranes of living suspension or adherent cells. Transfection and staining of MMSC membranes were carried out in accordance with our previous works [21,22,25,26].

### 2.2. Liver Resection and Introduction of MMSCs

A median laparotomy was performed, the caudal margin of the left lobe of the liver was resected, 100 μL of MMSC suspension (10^5^ cells) in warm physiological saline solution was injected into the injury site to a depth of 1–2 mm with an insulin syringe, and the anterior abdominal wall was sutured layer by layer. Intact animals, rats with liver resection without using MMSCs and after MMSC injection into the parenchyma of the intact liver were used as control. Animals with symptoms of pyoinflammatory processes were discarded and did not participate in further studies. For each point of the study, 8–12 rats were used (total 181 specimens). One, 2, 3, 4, and 5 weeks after the MMSC injection, the animals were sacrificed.

### 2.3. Morphological Research Methods

The left lobe of the liver was fixed in a 4% paraformaldehyde solution, dehydrated in an ethanol gradient of increasing concentration, clarified in xylene, and embedded in the histoplast. Sections with a thickness of 5–7 μm were stained with hematoxylin and eosin, to assess the number and distribution of macrophages on the sections, an indirect immunoperoxidase reaction with monoclonal antibodies against the CD68 antigen (Dako, Denmark) was carried out, studied under a light microscope Axioimager M1 (Zeiss, Germany) at magnification up to 1200 times.

In addition, unstained sections were examined in the luminescence mode of the indicated microscope with Alexa Fluor 488 or rhodamine (Rhod) filters. When obtaining micrographs, automatic exposure was used, in the process of combining images using the Alexa 488 and rhodamine filters, green and red (or orange and yellow) colors can be obtained, depending on the predominance of the glow intensity with a particular filter. Green was given by brighter fluorescence when using the Alexa 488 filter, red when using a rhodamine filter, yellow and its gradations are obtained by mixing green and red in one ratio or another.

The condition of the liver was examined near and far (at least 5 fields of view when using the X 10 objective) from the surgical site. Since there was no liver damage in the groups of intact animals and rats injected with MMSCs without resection, organ changes were assessed once in the center of the left lobe.

To determine the severity of neutrophilic infiltration, we measured images obtained with a digital video camera of a microscope on a computer screen using the software of the Axiovision morphological module (Zeiss, Germany). When using a lens with magnification of X 40, the area of the rectangular image was 8.7 × 10^4^ μm^2^ (sides 350 × 250 μm). Three to five measurements of different areas were performed on each preparation. During statistical processing of the obtained data, the arithmetic mean and standard deviation were determined. The significance of the difference between the compared mean values was determined on the basis of the Student’s test; the difference between the compared series with a confidence level of 95% and higher was considered significant. In the calculations, it was taken into account that the distribution of the studied characters was close to normal.

## 3. Research Results

At 1–3 weeks after liver resection, both without the use of cellular technologies and with the subsequent introduction of MMSCs, the forming scar (loose connective tissue) at the operation site was at first thick, infiltrated by macrophages, lymphocytes, and fibroblasts, then its thickness rapidly decreased, and their dense fibrous connective tissue appeared. Additionally, in the scar, very large (more than 50 μm) macrophages with foamy cytoplasm were found in abundance. Necrosis, areas with detritus, and numerous hemorrhages with leukocytes in the center and around were located in and under the scar (Figure 1a).

Gradually, the areas of necrosis and hemorrhages were reduced, the connective tissue structures located on the surface and deep in the liver became thinner. After 4 and 5 weeks, in most rats, the resection site was covered with dense fibrous connective and fibrous tissue, almost like a Glisson capsule. Connective tissue layers with groups of scleroid vessels and young bile capillaries departed from the operation site into the parenchyma (Figure 1b).

The number of neutrophils with segmented nuclei per unit area of the parenchyma section near the resection site only 5 weeks after surgery followed by injection of MMSCs (1.5 ± 0.756 cells per 10^5^ μm^2^ of the cut area) statistically did not significantly differ from the intact control (0.167 ± 0.389 leukocytes; *p* > 0.05) and was significantly less than for the same period in the group of animals after the same damage, but without the use of cellular technologies (4.2 ± 1.03 neutrophils; *p* < 0.05) (Figure 1c,d).

Necrosis and hemorrhages in the liver of rats after resection with MMSC injection disappeared by the 5th week after surgery, but after surgery without MMSC injection, hemorrhages and necrosis in some specimens at this time were found even at the last point of the experiment (Figure 1e,f).

According to luminescence microscopy, after the introduction of MMSCs into the resected liver in the scar and the restored capsule that developed at the site of resection, as well as in the layers of connective tissue extending deep into the parenchyma, there were many cells of different sizes, sometimes very large, up to 30 μm in size, with very intense glow when using a rhodamine filter. In such cells, the entire cytoplasm was not fluorescent but clearly outlined oval inclusions (Figure 2a).

The macrophage nature of such luminous objects was proved by immunohistochemical studies using monoclonal antibodies against the CD68 antigen; a large number of macrophages was shown in the developing scar. By the content of macrophages, even the border between the scar and the parenchyma of the organ can be clearly defined. Sometimes, almost all of the cells of the scar were macrophage cells (Figure 2b).

In the hepatic parenchyma distant from the surgical site, cells, located singly or in small groups, elongated along sinusoids with very bright fluorescence when installing a rhodamine filter, were found. Such cells were sometimes grouped near the vessels located on the periphery of the lobules (Figure 2c–f).

The number of cells glowing under the conditions of using the rhodamine filter decreased over time, and they completely disappeared from the liver of animals after the introduction of MMSCs into an intact organ by the 4th week. However, after the use of cellular technologies in the time of resection, cells fluorescent in this way remained in a fairly large number in the parenchyma and connective tissue structures during all 5 weeks of observation (Figure 2g). An immunohistochemical reaction with antibodies against the CD68 antigen showed a pronounced positive reaction in cells located along thin layers of connective tissue, in the same place where objects with predominant luminescence were found when using a rhodamine filter (Figure 2h).

In no case (8–12 rats for each point of the experiment, 3–5 sections from a liver sample from each animal) after the injection of MMSCs both into the intact liver and into the resection site was found the differentiation of the injected MMSCs into liver cells: whether hepatocytes or epithelial cells of the bile ducts and biliary tracts, or endotheliocytes and pericytes of the vessels, including granulation tissue, or even cells of connective tissue at the site of injury and in the parenchyma. This is evidenced by the absence of luminescence of both the GFP protein and Vybrant^®^ CM-Dil stained inclusions in these objects.

## 4. Discussion

There are many reports in the literature on the possibility of differentiation of externally introduced MSCs into hepatocytes [15,16] in the course of cell therapy for various pathologies of this organ. There are also results indicating the incorporation of MMSCs into the vessel walls of granulations developing at the site of tissues damaged during surgery [22]. Differentiation of MSCs into cells of connective tissue is possible [21,30].

According to our published data [21,22] and the results of other researchers [19,20,23,24], most of the MSCs injected into tissues remain viable for a very short time. A sharp change in the microenvironment of cell existence, the transition from comfortable conditions of in vitro cultivation to completely different conditions of in vivo functioning contribute to the death of a significant portion of the injected MSCs.

Upon destruction of the injected MMSCs, the Vybrant^®^ CM-DiI stained detritus is phagocytized by macrophages. This is evidenced by the bright fluorescence when using the rhodamine filter of many large cells of various shapes located in the structures of the scar at the site of liver damage. It is possible that the introduced MMSCs due to a certain foreignness (DNA of the GFP protein and the protein itself, viral transfection vector, some biochemical, and physiological differences, etc.) are the main reason for the increase in the number of macrophages (Kupffer cells).

The literature contains an article describing the foreignness of cells with the transfected GFP-gene [31]. In addition, when studying changes in the lymph nodes after cancer treatment, we found that with a decrease in the number of lymphocytes, monocytes, and neutrophils, this is compensated by an increase in the number of macrophages [32]. Therefore, it is possible that in cases of suppression of the myelocytic and lymphoid cells by MMSCs, this will also be compensated by macrophages.

Most likely, the data of other researchers on the possibility of differentiation of externally introduced MSCs into liver cells and structures [15,16] are somewhat exaggerated.

It is possible that the effect of MMSCs during liver resection is not based on direct participation in the repair processes, but on the secretion or release of certain cytokines during the destruction of cells, which have an immunomodulatory effect on the inflammatory reaction accompanying damage [17,18,33], which can somehow change its course and severity.

There is a possibility that MMSCs, their cytokines, and detritus activate and attract macrophages to the sites of their concentration [25,34,35]. As a result of more active phagocytosis of the debris, which occurs not only from the injected MMSCs, but also from the tissues destroyed during resection, the site of surgical intervention is cleared of antigenic substances faster, necrosis and hemorrhages disappear faster and earlier, the number of neutrophils normalizes, and reparative processes begin.

However, at the same time, it is possible that the decrease in the content of neutrophils with segmented nuclei and restoration of the organ structure is associated with the immunomodulatory effect of the injected MMSCs, which, either by themselves or through their exosomes, reduce the activity of acute inflammation, including in the liver [17,33], and also inhibit the functions of neutrophils [36,37].

It should be noted that there is a possibility of MMSCs getting directly into the bloodstream both during injection and in places of violation of the integrity of blood vessels during resection and as a result of the inflammatory process that developed in response to damage and injection of MMSCs [26]. Kupffer cells in areas of the organ distant from the MMSC injection site, located along the liver sinusoids and next to larger vessels, can capture the Vybrant^®^ CM-DiI stained MMSCs and their detritus from the bloodstream and, thus, begin to intensely luminesce when using a rhodamine filter. This assumption is supported by the presence of brightly fluorescent macrophages precisely at the sites of vascular branching, where the blood flow changes and where it is easier for macrophages to extract foreign substances from the blood—the same MMSC detritus.

Thus, on the basis of the foregoing, it can be concluded that the use of MMSCs in liver resection for the rapid restoration of an organ is inappropriate, since the introduced cells in vivo do not differentiate either into hepatocytes, into epithelial cells of bile capillaries, into endotheliocytes and pericytes of the vascular membranes, in fibroblasts of the scar or other connective tissue structures, or in any other cells present in the liver. It is possible that the effect of MMSCs is based not on direct participation in the processes of organ repair, but on the secretion or release cytokines during destruction that have an immunomodulatory effect on the inflammatory response accompanying damage.

## 5. Conclusions

The main features of the regeneration of the resected lobe of the liver under the conditions of using MMSCs are an earlier normalization of the number of neutrophils in the organ’s parenchyma and a more rapid disappearance of necrosis and hemorrhages.

After the introduction of MMSCs into both the intact and the resected liver, they do not differentiate into any cells of the recipient organism. In this case, the effect of MMSCs is based on the secretion or release cytokines during destruction, which have a pronounced immunomodulatory effect on the inflammatory response accompanying damage.

The bulk of MMSCs, after injection into the liver, is phagocytosed by Kupffer cells. Detritus of MMSCs from the liver partially enters the bloodstream, from where it is phagocytosed by paravasal macrophages.

## Figures and Tables

**Figure 1 life-11-00066-f001:**
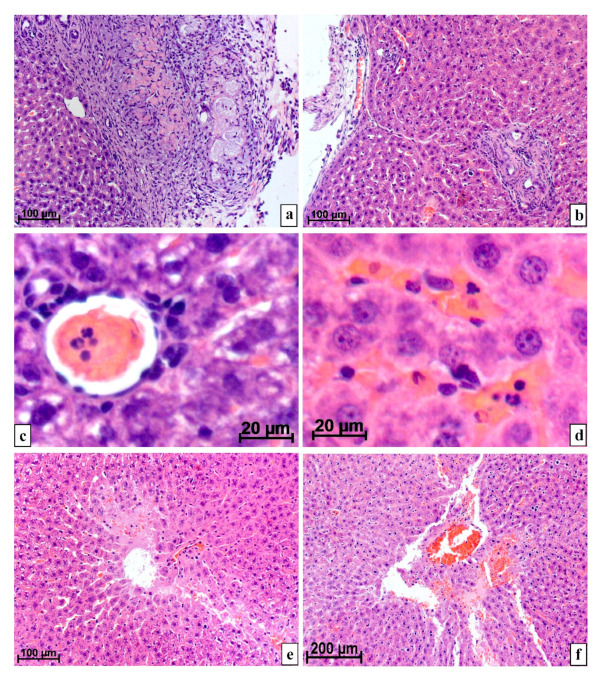
Liver of rats at various times after resection. Staining with hematoxylin and eosin: (**a**) After 1 week at the site of resection and injection of MMSCs, there is a wide forming scar extending deep into the parenchyma. The scar tissue contains many leukocytes, very large, more than 50 μm, cells with a foamy cytoplasm and structureless eosinophilic fragmented detritus; (**b**) a capsule of dense connective and fibrous tissue with a high content of macrophages was restored at the site of the operation with the subsequent introduction of MMSCs by 4th week. Thin layers of connective tissue containing groups of vessels and young bile capillaries depart from the scar on the surface into the parenchyma; (**c**) on the 2nd week after surgery, without MMSCs injection, some peripheral veins are thrombosed, there are many neutrophils in the thrombi; (**d**) after resection without the introduction of MMSCs, even after 5 weeks, there are many neutrophils in the parenchyma and vessels; (**e**) 4 weeks after damage and injection of MMSCs, areas of plasma impregnation with necrosis of hepatocytes remain in the parenchyma; (**f**) hemorrhage with plasma impregnation of the parenchyma and necrosis of adjacent cells on 4th week after resection and injection of MMSCs.

**Figure 2 life-11-00066-f002:**
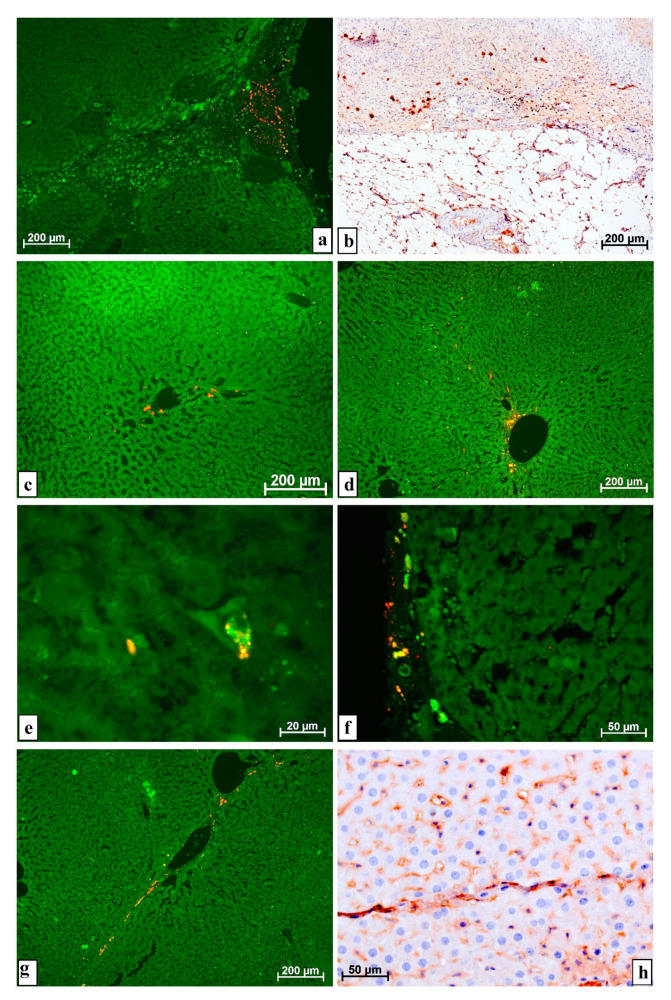
Results of a study using luminescence and immunohistochemistry of animal liver after resection and injection of MMSCs: (**a**) After 1 week in the developing scar on the surface of the organ, there are many objects with very bright fluorescence when using a rhodamine filter. In the layer of connective tissue that goes deep into the liver, there are no such objects. The result of computer alignment of images obtained using the Alexa 488 and rhodamine filters; (**b**) in the 1st week, the structures of the scar contain many cells of the macrophage series. There is a clear border between the macrophage scar and the preserved parenchyma. Reaction with antibodies against CD68 antigen, staining with diaminobenzidine and hematoxylin; (**c**) after 1 week, far from the operation site, near the peripheral vessels, there are groups of cells with very intense fluorescence when a rhodamine filter is installed. The result of computer alignment of images obtained using the Alexa 488 and rhodamine filters; (**d**) after 2 weeks, at a distance from the resection site, under the conditions of using a rhodamine filter, the cells located near a large vessel glow brightly. The result of computer alignment of images obtained using the Alexa 488 and rhodamine filters; (**e**) in the 3rd week, next to the bifurcated vessel, there is a large cell (about 20 microns) with a dark nucleus, some of its numerous cytoplasmic inclusions fluoresce brightly when using a rhodamine filter, while others do so when installing an Alexa 488 filter. The result of computer alignment of images obtained using Alexa 488 and rhodamine filters; (**f**) in the restored capsule on the surface of the organ, after 5 weeks, cells of various sizes and shapes intensively fluoresce in different colors. The result of computer alignment of images obtained using the Alexa 488 and rhodamine filters; (**g**) a thin layer of connective tissue at a distance from the operation site in the 5th week contains wide vessels and cells with brighter fluorescence against the background of the application of a rhodamine filter. The result of computer alignment of images obtained using the Alexa 488 and rhodamine filters; (**h**) after 5 weeks, a pronounced positive reaction to the histiocytic antigen was noted in the cells of a thin layer of connective tissue at a distance from the site of damage. Macrophages are located in one row and sometimes completely surround other cells that are not hepatocytes. Reaction with antibodies against CD68 antigen, staining with diaminobenzidine and hematoxylin.

## Data Availability

The data presented in this study are available on request from the corresponding author.

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
