# Peer review of "Some Special Aspects of Liver Repair after Resection and Administration of Multipotent Stromal Cells in Experiment"

_life, 2021, doi:10.3390/life11010066_

Round 1

Reviewer 1 Report

Maiborodin et al. have tested the effect of MSCs transplantation in liver repair and regeneration. The study aim is sound, but the study outcome can be improved. Some comments for authors are:

  1. The title is misleading.

  2. Please check the manuscript for typos.

  3. Some positive control is required to show that the MSCs transplantation procedure does not result in a significant loss of MSCs required for differentiation. Further, the differentiation potential of MSCs is maintained (like including in vivo or in vitro studies to support the observation).

  4. Instead of differentiation, literature reports stimulation of existing hepatocytes etc. on MSCs transplantation in the liver injury site. Comment

Author Response

I thank the reviewer for studying our manuscript and for the comments made

  1. The title is misleading.

The manuscript was initially titled: "Lack of differentiation of multipotent stromal cells after administration into resected liver", but this name did not reflect changes in tissue infiltration by neutrophils and macrophages, so the name was changed to the present. I agree that this name is not ideal and does not reflect the lack of differentiation of multipotent stromal cells to hepatocytes, but the obtained data are presented in more detail in the abstract. We will be glad if the Dear Reviewer suggests a more optimal title.

  1. Please check the manuscript for typos.

Several typos were found. Errors corrected.

  1. Some positive control is required to show that the MSCs transplantation procedure does not result in a significant loss of MSCs required for differentiation. Further, the differentiation potential of MSCs is maintained (like including in vivo or in vitro studies to support the observation).

MSCs, in preparation for the experiment, were tested for the possibility of differentiation. The capacity of induced in vitro differentiation into osteogenic, adipogenic and chondrogenic lineages is the only critical requirement for choosing MSCs population. MSCs can differentiate into bone tissue cells and the conditions of this differentiation can be easily reproduced, therefore this differentiation is routinely used for in vitro characterization of MSCs cultures and is a typical default differentiation pathway for the majority of MSCs in culture. Osteogenic differentiation was induced by adding 0.1 μM desoxymethasone, 50 μM ascorbic acid, and 10 μM β-glycerophosphate (Sigma) (Shima W.N. et al., 2015; Qiu K. et al., 2016; Chen Y. et al., 2018). Osteogenic differentiation was determined by two markers: activity of alkaline phosphatase and mineralization of extracellular matrix with calcium ions. Cytochemically, alkaline phosphatase was detected using nitroblue tetrasolium in the presence of 5-bromo-4-chloro-3-indolyl phosphate as the substrate. Accumulation of calcium in extracellular matrix was recorded by alizarin red staining.

The section "Material and Methods" is supplemented.

Unfortunately, MSCs in tissues, especially damaged ones, are very quickly destroyed and phagocytosed by macrophages. This is due to both a sharp change in the conditions of existence, and the unfavorable influence of the wound separable. We mentioned this in the manuscript:

«Along with this, there is evidence in the literature that after injection into tissues the MSCs die very quickly due to a sharp change in the conditions of existence [19-22].»

«According to published our data [19,20] and the results of other researchers [21,22], most of the MSCs injected into tissues remain viable for a very short time. A sharp change in the conditions of cell existence, the transition from comfortable conditions of in vitro cultivation to completely different conditions of in vivo functioning contribute to the death of a significant portion of the injected MSCs.»

We do not think that the differentiation potential of MSCs is preserved, we describe the lack of differentiation in tissues in this manuscript.

  1. Instead of differentiation, literature reports stimulation of existing hepatocytes etc. on MSCs transplantation in the liver injury site. Comment

I am very critical of the possibility of stimulating of existing hepatocytes by MSCs, but I do not exclude this possibility. However, based on the suppression of myelocytic and lymphoid cells by MSCs, I consider the likelihood of hepatocyte stimulation to be insufficiently studied.

I hope that we have responded to all the comments and recommendations of the distinguished reviewer, and I thank him again for the work done.

Changes to the text are highlighted in color.

Reviewer 2 Report

The manuscript by Maiborodin et. al describes the effects of mesenchymal stem cells (MSCs) from bone marrow during liver regeneration. They observe no differentiation of MSCs in the scar, however changes in the inflammatory cell subsets, in particular of macrophages and neutrophils, were detected.  However, the main point of this conclusion is not fully documented. This is an interesting point, indeedit has been recently described that MSC reduces liver damage by suppressing neutrophil and macrophage recruitments (Wan et al. PLoS One 2020 Feb 11;15(2):e0228889.).

In line 240 authors state that: “It is possible that the introduced MMSCs due to a certain foreignness (DNA of the GFP protein and the protein itself; viral transfection vector; some  biochemical and physiological differences; etc.) are the main reason for the increase in the number of macrophages (Kupffer cells).”

However, the appropriate control is missing. Indeed injection of the parental unlabeled MSCs is required to understand if macrophages recruitment is an artefact due to MSC manipulation or it is a consequence of changes in cytokines/chemokines secretion after MSC injection.  

Minor comments:

Abstract. The first sentence of the abstract should be rephrased. The mention to a commercial product should be avoided in the abstract, I suggest removing it.  

Throughout the manuscript “;” was used instead of “,”. The authors should correct the punctuation.

Lines 55. “in the microenvironment” seems be more appropriate than “in the condition of existence”.

Line 95. Maybe the authors mean that the animals were “sacrificed”, “withdrawn” seems inappropriate.

Author Response

I am grateful to the reviewer for studying our manuscript and for the time spent.

In line 240 authors state that: “It is possible that the introduced MMSCs due to a certain foreignness (DNA of the GFP protein and the protein itself; viral transfection vector; some biochemical and physiological differences; etc.) are the main reason for the increase in the number of macrophages (Kupffer cells).”

However, the appropriate control is missing. Indeed injection of the parental unlabeled MSCs is required to understand if macrophages recruitment is an artefact due to MSC manipulation or it is a consequence of changes in cytokines/chemokines secretion after MSC injection.

In this case we do not claim authorship of the idea that “introduced MMSCs due to a certain foreignness are the main reason for the increase in the number of macrophages”. This is only an attempt to theoretically explain the possible reason for the increase in the number of macrophages. Perhaps we will further investigate this phenomenon, or someone else will do it.

At the same time, the literature contains an article describing the foreignness of cells with the transfected GFP-gene (Rosenzweig M., Connole M., Glickman R., Yue SP, Noren B., DeMaria M., Johnson RP Induction of cytotoxic T lymphocyte and antibody responses to enhanced green fluorescent protein following transplantation of transduced CD34 (+) hematopoietic cells. Blood. 2001; 97 (7): 1951-9.). In addition, when studying changes in the lymph nodes after cancer treatment, we found that with a decrease in the number of lymphocytes, monocytes and neutrophils, this is compensated by an increase in the number of macrophages (Kolotova NM, Maiborodin IV, Fursov SA, Lushnikova EL, Zarubenkov OA, Maiborodina VI. Morphology of pararectal lymph nodes in rectal cancer after neoadjuvant therapy. Bull Exp Biol Med. 2010; 149 (2): 250-4. doi: 10.1007/s10517-010-0919-y.). Therefore, it is possible that in case of suppression of the myelocytic and lymphoid cells by MMSCs, this will also be compensated by macrophages.

Links to the indicated authors have been added to the manuscript

Abstract. The first sentence of the abstract should be rephrased. The mention to a commercial product should be avoided in the abstract, I suggest removing it.

Product commercial name changed

Throughout the manuscript “;” was used instead of “,”. The authors should correct the punctuation.

I apologize. Somewhere in the process of preparing the manuscript, we accidentally made an autocorrect. Fixed

Lines 55. “in the microenvironment” seems be more appropriate than “in the condition of existence”.

Changed

Line 95. Maybe the authors mean that the animals were “sacrificed”, “withdrawn” seems inappropriate.

Changed

I once again want to thank the distinguished reviewer for the clear and specific recommendations and for the work spent studying our manuscript.

Changes to the text are highlighted in color.

Round 2

Reviewer 1 Report

All the comments have been addressed by the authors. Refarding the title of the manuscript, I would suggest something on lines of, "Failure/lack of liver repair after resection and administration of Multipotent stromal cells".

Reviewer 2 Report

The authors have accomplished the requests.